# Why managing sciatica is difficult: patients' experiences of an NHS sciatica pathway. A qualitative, interpretative study

Clare Ryan  ,[1,2] Catherine J Pope,[3] Lisa Roberts[2,4]

¹Physiotherapy Service, Solent NHS Trust, Portsmouth, UK
²School of Health Sciences, University of Southampton, Southampton, UK
³Nuffield Department of Primary Care Health Sciences, University of Oxford, Oxford, UK
⁴Therapy Services Department, University Hospital Southampton NHS Foundation Trust, Southampton, UK

**Correspondence to**
Clare Ryan;
Clare.ryan1@nhs.net

## ABSTRACT

**Objectives** Amid a political agenda for integrated, high-value care, the UK is implementing its Low Back and Radicular Pain Pathway. To align care with need, it is imperative to understand the patients' perspective. The purpose of this study was, therefore, to explore how people experience being managed for sciatica within an National Health Service (NHS) pathway.

**Design** Qualitative interpretative study.

**Setting** Musculoskeletal Service in an NHS, Primary Care Trust, UK.

**Participants** The sample comprised 14 people aged ≥18 years with a clinical presentation of sciatica, who were currently under the care of a specialist physiotherapist (the specialist spinal triage practitioner), had undergone investigations (MRI) and received the results within the past 6 weeks. People were excluded if they had previously undergone spinal surgery or if the suspected cause of symptoms was cauda equina syndrome or sinister pathology. Participants were sampled purposively for variation in age and gender. Data were collected using individual semi-structured interviews (duration: 38–117 min; median: 82.6 min), which were audio-recorded and transcribed verbatim. Data were analysed thematically.

**Results** A series of problems with the local pathway (insufficient transparency and information; clinician-led decisions; standardised management; restricted access to specialist care; and a lack of collaboration between services) made it difficult for patients to access the management they perceived necessary. Patients were therefore required to be independent and proactive or have agency. This was, however, difficult to achieve (due to the impact of sciatica and because patients lacked the necessary skills, funds and support) and together with the pathway issues, this negated patients' capability to manage sciatica.

**Conclusions** This novel paper explores how patients experience the process of being managed within a sciatica pathway. While highlighting the need to align with recommended best practice, it shows the need to be more person-centred and to support and empower patient agency.

**Trial registration number** ClinicalTrials.gov reference (UOS-2307-CR); Pre-results.

### Strengths and limitations of this study

► To our knowledge, this paper is unique in exploring how patients with sciatica experience the process of being managed within a pathway.

► Although this is a small, single-site study, it is rich in detail and has conceptual transferability to similar services at an early stage of pathway implementation managing patients who have not improved with conservative treatment and whose symptoms are sufficient to require investigation.

► Limitations include not undertaking iterative theoretical sampling during analysis and interviewing some participants before they had completed their management.

## INTRODUCTION

Sciatica is characterised by pain, altered sensation and/or weakness in the leg. It is thought to result from compression, inflammation and/or sensitisation of a lumbar/sacral nerve root, caused by lumbar disc herniation and stenosis, or rarely, sinister pathology (such as cancer); however, it is not always possible to identify a structural cause.[1 2] Sciatica differs from somatic referred leg pain that is attributed to structures other than the nerve root, such as the joints, muscles and ligaments. It is well-established that the presence of sciatica, compared with low back pain alone, or somatic referred leg pain, increases symptom severity, disability, absence from work and negatively impacts on health outcome.[3] Furthermore, patients report that sciatica can be incapacitating and severely limit daily activities.[4] Sciatica prevalence estimates vary widely, from 1.2% to 43%, reflecting differing diagnostic criteria and sampling methods.[5] While the majority of patients with sciatica experience early improvement in symptoms, usually in the first 2–3 months, either with or without treatment; a minority will experience more persistent

symptoms or disability, and for some this continues beyond 12 months. Some patients will experience intermittent or recurrent sciatic symptoms over time. There is, however, inconsistency in the literature about the proportion of patients affected by ongoing symptoms. While one review found this to be as few as 13% of patients at 6 months,[6] another study based in primary care, found that for 45% of patients, disability had failed to significantly improve at 12 months.[7] The reason for this variation is not fully understood but may be associated with symptom duration at presentation or the type of studies to which patients were recruited (clinical vs epidemiological). Recent work indicates that factors negatively associated with recovery include longer leg pain duration; more symptoms associated with sciatica; and patients' belief that the problem would last a long time; conversely, having myotomal weakness was positively associated with recovery.[7]

Over the past decade, in the UK and the developed world, there has been a political agenda for healthcare that is integrated (collaborative and responsive) and high value (aligns with best practice).[8] Reflecting these agendas, and to facilitate implementation of recent National Institute of Health and Care Excellence guidance,[9] a UK Low Back and Radicular Pain Pathway[10] has been developed. Achieving integrated, high-value care is particularly important for patients with sciatica who commonly cross sectors of care to access management from a range of health professionals. The new national pathway recommends that patients with sciatica are initially managed within primary care, with medication and/or physiotherapy, with a 2-week review after initial presentation. For those with severe pain that fails to improve, assessment by a specialist (spinal triage) practitioner is recommended at 2–6 weeks, or at 6 weeks for those with non-tolerable pain. In the UK, this specialist role is commonly performed by an advanced practice physiotherapist.[10] The pathway recommends that investigations, such as MRI, are offered (if they are likely to change management) and completed within 4 weeks, with results review within 2 weeks. Where imaging supports the suspected clinical diagnosis of sciatica, referral for an epidural injection or surgery is advised, with surgery occurring within 8–12 weeks of the onset of symptoms, or for those with very severe symptoms, within 3 weeks. When no concordant structural cause is identified, or the patient wishes to avoid invasive intervention, the recommended approach is a combined physical and psychological programme. The pathway advocates that clinicians share decision-making with patients at each stage of the pathway.

Understanding patients' experiences of being managed for sciatica can inform the provision of care. Four qualitative UK-based studies[11–14] with a total of 75 patients have provided some insight into patients' beliefs, expectations and experiences of components of the conservative management of sciatica. These papers do not, however, provide an overview of what it is like to be managed in a sciatica pathway. This paper therefore addresses the question, 'what are patients' experiences of being managed within an National Health Service (NHS) sciatica pathway?'

## METHODS
The data for this paper were collected as part of a wider study about patients' experiences of investigations for sciatica.[13] This paper is reported in line with the consolidated criteria for reporting qualitative research (COREQ) guidelines.

### Study design
This was an interpretative, qualitative research study, a methodology consistent with the aim of exploring peoples' experiences.

### Setting
The setting was an outpatient physiotherapy service of a UK, NHS, Community Trust. This service was selected as it was working to align its practice with the new national pathway and had experienced specialist spinal triage practitioners. At the time data were collected, the first version of the national pathway had been available for 16 months. Over this period, changes had occurred in the spinal surgery, injection and imaging providers. Not all providers were commissioned and funded by the same organisation. Issues with staff shortages and provider negotiations had resulted in waiting times of 6 months for a routine appointment with a spinal surgeon and 4 months for an injection (however, patients could choose to access this in an adjacent city within 4 weeks). Following a recent change in provider, the wait for imaging had reduced from 6 to 3 weeks.

### Participants and recruitment
The sample comprised people aged ≥18 years with a clinical presentation of sciatica, who were currently under the care of a specialist physiotherapist (the specialist spinal triage practitioner), had undergone investigations (MRI) and received the results within the past 6 weeks. Participants were, therefore, recruited at a key point in their pathway management, when they had failed initial conservative management, had undergone investigations to determine whether a concordant structural cause could be identified, and had discussed with a specialist physiotherapist the most appropriate next step.

The study was designed to recruit 12–15 participants. No claims were made to reach data saturation, as each person's experience of a phenomenon is unique. However, aligning with qualitative research recommendations,[15 16] this number was considered enough to enable a rich, detailed analysis of this defined, relatively homogenous group, while providing sufficient information to answer the research question, and sufficient variation within the sample to enhance transferability.

The clinical diagnosis of sciatica was made by the patient's specialist physiotherapist using information from their clinical assessment and based on diagnostic criteria (figure 1) reflecting those used in practice.[17] The patient's radiological findings did not inform this diagnosis. The specialist physiotherapists were clinicians with ≥10 years musculoskeletal experience, who undertook specialist spinal training ≥four times a year. We used

Leg pain and *at least one* of the following (on the affected leg): sensory deficit (to touch or pinprick); muscle weakness; impaired tendon reflex; a positive straight leg raise reproducing the person's familiar pain; or a positive prone knee bend test combined with anterior thigh pain.

**Figure 1** Criteria for diagnosing sciatica.[17]

purposive sampling to gain representation across age, sex and duration of symptoms, and recruited 14 participants, sufficient to enable rich, detailed analysis.

Potential participants were approached about participating in the study, by their specialist physiotherapist, when they attended for their investigation results. Those interested were provided with verbal and written information about the study. With their written consent, the researcher made contact to arrange an interview date. To increase the homogeneity of the sample, people were excluded if they had previously undergone spinal surgery or the suspected cause of symptoms was sinister pathology or cauda equina syndrome. Patients were also excluded if they were unable to communicate without the assistance of an interpreter; they lacked capacity to provide consent; or the researcher had treated them in a previous episode of back pain (to minimise bias).

### Data collection

The lead author (CR) a female specialist physiotherapist and MRes student (with prior experience and training in conducting qualitative interviews) collected data between October 2015 and May 2016 using in-depth, individual, face-to-face, semi-structured interviews. This method was selected to facilitate rapport and enable emergent issues to be explored. To minimise the influence of the researcher's position, CR introduced herself as a researcher, and hosted the interviews in an individual room within the hospital, away from the physiotherapy department.

1. When and how did you first seek help for your leg pain?
   Probes: Who initiated, why, who did you see?

2. Who has treated you for leg pain?
   Probes: GP; Physio; physio specialist; others; how long with each; roles; coordination of care?

3. How were management decisions made?
   Probes: shared decisions?

4. How did you move between the various clinicians?
   Probes: who initiated; shared decision?

5. How have you managed your leg pain yourself?
   Probes: what helps, strategies tried

6. What aspects of your management have been most and least helpful?
   Probes: positive/ not so, what was most important/ useful

**Figure 2** Topic guide.

A topic guide was used as a starting point for discussion, however, interviews followed the participant's lead, with emergent issues being explored and, where indicated, incorporated into subsequent interviews. Figure 2 details the key questions used to explore patients' experience of the pathway. Minimal facilitation was used to prevent 'leading'. At the start of data collection, CR conducted pilot interviews with two participants to provide face and content validity for the questions asked. As the content and key wording of questions were substantially unchanged, the data from these participants were included in the study. The interviews were audio-recorded, transcribed verbatim and pseudonyms were used to maintain anonymity. We did not ask participants to validate the transcripts or findings as the usefulness of this strategy is contested.[18]

### Patient and public involvement

Informal discussions with a series of patients, members of the Southampton branch of BackCare, and engagement with the public during three-dissemination events, aided the interpretation of findings.

### Data analysis

We analysed the data manually, thematically and iteratively, based on the method of Braun and Clarke.[19] Thematic analysis has three stages: (1) line-by-line coding, (2) developing descriptive themes and (3) generating analytical themes. First, we explored the data on its own terms, using inductive line-by-line coding. The data relevant to the research question were identified, explored through selective and axial coding, and initial concepts identified. We then explored relationships and potential explanations for concepts, moving back and forth between the raw data, key concepts and relevant clinical, theoretical and policy literature. We used charts to manage the data, to facilitate comparison within and between cases and to ensure that analysis remained rooted in the data.[20] We included variation and complexity and used analytical and reflexive memos to facilitate a deeper understanding.[20] Each of the authors contributed to analysis; CR identified the initial codes and concepts, which CP and LR, expert qualitative researchers, interrogated to refine their scope; relevance; constituent parts; relationships; and explanatory potential. Involving three analysts with different professional backgrounds aided the rigour of analysis and provided a check to minimise potential bias, assumptions and data selection.

### RESULTS

The sample comprised 14 participants; aged 34–81 years (median 61 years); 8 participants were women. Participants described a 3-month to 9-year (median 13 months) duration of symptoms. By the time of their interview, half of participants had experienced improvement in their symptoms, including five with significant improvement. Participant characteristics are detailed in table 1. Four additional people were identified but not included as

**Table 1** Participant characteristics[4 13]

| Participant, age and sex | Work status and occupation | Symptoms | Symptom duration | Neurological findings | Pathway management for this episode of sciatica | MRI findings* | Likely next step |
|---|---|---|---|---|---|---|---|
| Julia 63 years Female | Part-time office worker | Leg pain, altered sensation, giving way and antalgic gait | 3 years | Positive ipsilateral SLR and impaired sensation power or reflexes | GP, physio, podiatry, private chiropractor and specialist physiotherapist | 1 | Nerve root block |
| Catherine 60 years, Female | Unemployed shop worker | Leg pain, altered sensation and cramp; back ache (leg pain>back) and difficulty weight bearing | 11.5 months | ? Positive ipsilateral SLR (unclear data) | GP, physio, private physio and specialist physiotherapist | 2 | Self-management with review |
| David 74 years Male | Retired professional | Leg pain>back pain and difficulty weight bearing | 3 months | Impaired sensation power or reflexes | GP, physio, private physio and specialist physiotherapist | 1 | Self-management |
| John 34 years Male | Unemployed delivery driver | Leg pain | 9 months | Positive ipsilateral SLR | GP, physio, private physio, specialist physiotherapist | 3 | Primary care physio |
| Daniel 37 years Male | Shop worker: on sick leave | Leg and back pain; back spasm, leg giving way, saddle anaesthesia | 21 months | Positive ipsilateral SLR and impaired sensation power or reflexes | GP and physio, specialist physiotherapist and two nerve root blocks (at pain clinic) | 2 | Pain management programme |
| Janet 73 years Female | Retired | Leg pain back pain, leg giving way, foot numb, difficulty weight bearing | 7 months | Positive ipsilateral SLR | GP, physio, private physio and specialist physiotherapist | 1 | Nerve root block |
| Bill 61 years Male | Part-time manual worker | Leg pain, back pain, altered sensation in legs, leg giving way | 10 months | Positive ipsilateral SLR and impaired sensation power or reflexes | GP and specialist physiotherapist | 1 | †MDT review |
| Claire 45 years Female | Unemployed | Leg pain>back pain, altered sensation leg and foot | 15 months | Impaired sensation power or reflexes | GP, physio and specialist physiotherapist | 2 | Nerve root block |
| Ruth 74 years Female | Retired shop worker | Leg pain | 3 months | Positive ipsilateral SLR | GP, physio and specialist physiotherapist | 1 | Physio (Unclear why nerve root block not offered) |
| Henry 81 years Male | Retired manual worker | Leg pain and difficulty weight bearing | 8 months | Positive ipsilateral SLR | GP, physio and specialist physiotherapist | 3 | Review appt with spinal specialist |
| Frances 72 years Male | Retired office worker | Leg pain and altered sensation in leg and foot | 18 months–2 years | Impaired sensation power or reflexes | GP, physio and spinal specialist. Specialist physiotherapist and surgical opinion in last flare-up. | 1 | †MDT review |
| Aisha 35 years Female | Office worker; currently off sick | Back and leg pain and altered sensation in legs | 6 years (6 month this episode) | Positive ipsilateral SLR | GP, physio and spinal specialist | 1 | Surgical opinion |
| Gareth 45 years Male | Office and manual worker; on light duties | Leg>back pain and altered sensation and spasm in legs and feet | 7 years | Positive ipsilateral SLR | GP, physio and specialist physiotherapist | 2 | Offered but declined pain management |
| Joanne 46 years Female | Housewife | Back and leg pain, altered sensation in legs and foot, saddle anaesthesia | 3 years | Positive ipsilateral SLR | GP, physio and spinal specialist. Previous visit to spinal specialist. Pain management programme | 2 | Physio |

GP: General Practitioner SLR: straight leg raise
*MRI results' categories: 1 Consistent with sciatica of nerve root origin. 2: Potentially relevant to symptoms but not consistent with sciatica of nerve root origin. 3: Do not appear relevant to patient's symptoms.
†Multidisciplinary team (MDT) review: meeting with pain clinic consultant and orthopaedic surgeon to ascertain.

they did not meet inclusion criteria (n=1); did not wish to participate (n=1); attended on the wrong day (n=1); or could not be contacted (n=1). Interviews ranged in length from 38 to 117 min (median 82.6 min).

Our analysis explored how people experienced being managed for sciatica within an NHS pathway. We focused on the issues patients perceived mattered most and which had important implications for policy and practice. We present the findings under three thematic headings: (1) problems with the pathway, (2) required agency and (3) the burden of agency. The term 'agency' refers to the independent and proactive actions undertaken by a patient as part of the self-management of their health.[21] We identify a series of problems with the pathway and

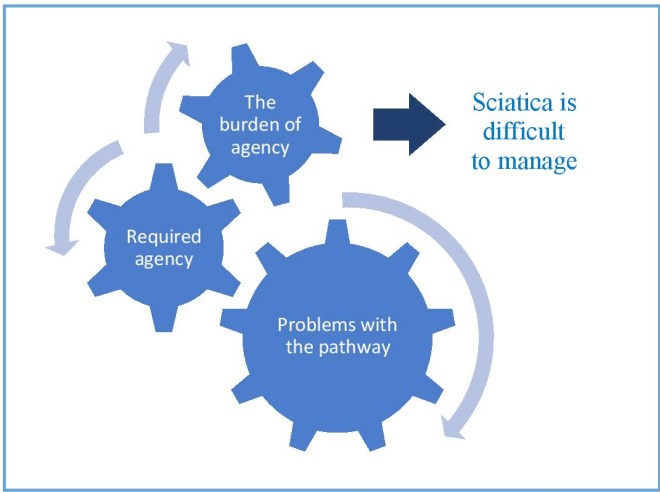

**Figure 3** Patients' experience of the pathway.

show that they required patients to be independent and proactive, or have agency, to meet their healthcare needs. Being independent and proactive was, however, difficult and burdensome for patients to achieve, and together with the pathway issues, negated patients' capability to manage sciatica. Patients' experience of the pathway is illustrated in figure 3.

### Theme 1: problems with the pathway

Our data showed that a series of problems with the pathway (listed in figure 4) made it difficult for patients to access the management they perceived necessary. Participants reported that neither the way the pathway worked nor the services available to them, including the risks, benefits and alternatives, were transparent.

> He didn't tell me the procedure, I don't think he [GP] knows the procedure …you won't find it anywhere, it's only when you experience it that you know how it works. (Henry 81 years)

Also, it was difficult for patients to contribute to management decisions because they were, by default, clinician-led or paternalistic.

> I had no control over it [the next step], it was 'we're going to try this, we're going to do this.' (Daniel 37 years)

Some participants reported that their specific needs and circumstances were not heard; some attributed this to care being protocol driven.

- Insufficient transparency and information
- Clinician-led decisions
- Standardised care
- Restricted access to specialist care
- A lack of collaboration between services

**Figure 4** Problems with the pathway.

I just wanted to be heard, wanted them to actually listen … I feel like people with sciatica pains they are branded with the same stick…I'm sure the person next door with sciatica … would be given the exact kind of thing to do. (Aisha 35 years)

> She [physio] was just doing what she was told to do … she gave me five of the blueprint [exercise protocol] of what she had for my case. (Daniel 37 years)

Accessing a specialist opinion (from either the specialist physiotherapist, or a surgical or pain clinic consultant) and investigations were reported to be difficult and protracted. This was because of the gatekeeping performed by their GP, physiotherapist and specialist physiotherapist, and having to first attend and fail physiotherapy.

> I said 'Dr X [GP] suggested that actually it might be a good idea for you to refer me on to see a neuro-consultant … And she [spinal specialist] … said 'Let me just stop you there… to be honest with you the neuro-consultant… they won't be interested in this at all'. (Gareth 45 years)

Finally, pathway services appeared to be compartmentalised and non-collaborative. Apart from the multidisciplinary team meeting, where the specialist physiotherapist and consultant collaborated, services appeared to function with limited co-operation, leaving patients to make sense of conflicting opinions and to join-up their own care. They were also required to join a new waiting list at each stage of the pathway, which resulted in recurrent delays.

> She [GP] hasn't done a full [sickness] certificate because she said I have nothing on my screen to show the new results from the MRI… all this toing and froing adds to the stress. (Catherine, 60 years)

> They are sending me to the pain clinic, but I'm told there's an 18 week list… so I'm not looking at any quick fix. (Janet 73 years)

Collectively, these problems compromised the extent to which sciatica management was enabling; personalised; collaborative and involved patients; the result was a pathway that was insufficiently person-centred.

### Theme 2: required agency

Our data showed that to manage sciatica, patients were required to be independent and proactive. As well as being required to navigate the problems with the pathway, all participants reported that they independently sourced resources from outside of the NHS. They accessed factual information (through webpages or newspaper articles); the experiences of others who had experienced sciatica; and private treatment. Patients perceived these resources to be necessary to make sense of their management options, gain prompt access to treatment or try options otherwise unavailable. Several participants also considered accessing surgery abroad, because it was quicker

and easier to access than NHS surgery, and cheaper than private treatment in the UK.

> I googled to try and find a direction in 'Well this is what's gone wrong, this is how you sort it out.' Or even a clue to say 'Fine, go to see your GP, try and point them in this direction'. (Gareth 45 years)

> [An acquaintance] went abroad because he felt like nothing was being done in this country soon enough…. [and] because it was cheaper. (Aisha 35 years)

Patients perceived that agency was required of a 'good' patient, illustrated in their accounts of being proactive, positive and compliant.

> I've looked into Pilates classes as well … try and do anything that's going to ease it or help. (Claire, 45 years)

> The doctor suggest to me as well that I need to walk a lot. That's why I try my best you know to walk more. (John, 34 years)

### Theme 3: the burden of agency

Finally, being independent and proactive was difficult and burdensome. Participants reported that the impact of sciatica negated their ability to make decisions; to adhere to prescribed management; to navigate the pathway; and remain positive and motivated to self-manage.

> It reduces me to tears and at times and I can't cope with that in my own mind somehow. I feel like I'm letting myself down by allowing it to take over. (Ruth, 74 years)

> I didn't want to feel a failure… I'm trying so hard to help myself, but it [exercise programme] was making it worse. (Joanne, 46 years)

Participants also reported lacking the required skills to access and evaluate information. Some were unaware of the need to critically appraise information and at times used information misaligned with the evidence to inform their management preferences.

> I got a lot of the information from the internet, now I'm rubbish on the internet, somebody else got it for me…people who've had it, and what they've had done, and it's got feedback and everything else. So, all of this leads to the options that I just told you. (Henry 81 years)

Some reported lacking the financial resources to access to private treatment, especially if multiple treatments, or costly interventions such as injections or surgery, were needed.

> I don't have any money to go back there [abroad] for an operation. (John, 34 years)

Finally, some participants lacked the support necessary for agency.

> I just went into physio, just did what I had to do. It was like I'd given up. (Catherine, 60 years)

## DISCUSSION

To our knowledge, this paper is unique in exploring how patients experience the process of being managed within a sciatica pathway. The findings are timely as NHS England implements its pathway and as other countries seek to understand the potential benefits[22]; and because of the recent drive in health policy for integrated, high-value care.[8 23] Although this is a small study, it is rich in detail and has conceptual transferability to similar services that manage patients who have not improved with conservative treatment and whose symptoms are sufficient to require investigation. Limitations include not undertaking iterative theoretical sampling during analysis (because analysis for this paper was completed after data collection had ended) and interviewing some participants before they had completed their management (to align with the aim of the original study, patients were interviewed within 6 weeks of undergoing investigations).

We identified three key themes about patients' experiences of being managed within a sciatica pathway: (1) problems with the pathway, (2) required agency and (3) the burden of agency. We found that problems with the local sciatica pathway compromised patients' ability to access the management they perceived necessary. These included insufficient transparency and information; clinician-led decisions; restricted access to specialist care; and a lack of collaboration between services. These issues have previously been noted in the sciatica and low back pain literature.[11 14 24–26] They are also already recognised in the national pathway as important issues to address. We suggest that although the local pathway focused on delivering the 'mechanics' of the pathway (procuring services; enabling faster access to imaging; and setting up regular case discussion meetings), it did not prioritise ensuring that management was person-centred. Our data reveal a culture of paternalism, protocol-driven care, information being provided on a 'need to know basis' and siloed services. Patients' frustration with the lack of timely access to a specialist opinion, likely reflects in part, the known issues with staff shortages and difficulty negotiating provider contracts. However, in addition, the national pathway places a new emphasis on enabling rapid access to imaging and injection or surgery (for those with severe or intolerable symptoms and a relevant cause identified on imaging) and achieving this requires a shift in practice and resources. While this approach reflects expert opinion, it is in contrast to the established 'stepped model' of sciatica care that advocates the more limited use of invasive interventions, with evidenced cost effectiveness.[27] We suggest that until the national pathway is underpinned by evidence of its clinical and cost effectiveness, services may be ambivalent about making the changes necessary to align with its timelines. While these

issues are specific to the local pathway, they seem likely to apply to other services in the early stages of pathway implementation.

Aligning with previous studies,[11 14] we found thatto manage sciatica, patients independently accessed information; the experiences of others; and private healthcare. They also perceived that a 'good' patient was necessarily independent and proactive. While new to the sciatica literature, these findings align with the experiences of those self-managing long-term conditions.[25] This paper furthers understanding by reframing patients' actions as agency, and arguing that this agency becomes necessary in part due to the problems with the pathway as well as in response to wider neo-liberal expectations that people manage their own health and reduce dependence on the state.[28] This study shows that it has become normal for both patients and staff to expect patients to be proactive and independent in their management. However, our study also found that the requirement to be independent and proactive when managing sciatica was difficult and burdensome for patients. This was because of the impact of sciatica and patients' lack of skill, funds and support. The Burden of Treatment Theory recognises that when patients are expected to participate in their healthcare but lack adequate support, this adversely affects their ability to manage their illness.[29] Our analysis demonstrates the particular burden of being required to navigate the pathway issues and to independently and proactively manage sciatica without the necessary support.

## Implications
Local implications include the need to improve and then re-evaluate the following aspects of care: patient information about the pathway; staff training and patient resources to enable effective shared decision-making; collaborative, joined up working; and timely progression through the pathway. Also, patients' experiences should be routinely explored as part of service evaluation. Transferable implications include the need to ensure pathway care is personcentred and provides the support and resources necessary to enable and empower patients to manage sciatica independently and proactively.

Further research exploring patients' experiences of the national pathway, particularly in services in which the new pathway is fully implemented, would enable a more comprehensive understanding of patients' experience of being managed in NHS sciatica pathways.

## CONCLUSION
A series of problems with the local pathway made it difficult for patients to access the management they perceived necessary. Patients were required to be independent and proactive in their management, but this was difficult to achieve and burdensome, due to sciatica symptoms, and lack of skills, funds and support. Both the pathway issues and the requirement to be independent and proactive negated patients' capability to manage sciatica. The key

implications are the need for sciatica pathways to align with recommended best practice, and also to be more person-centred and to empower and enable patients to manage sciatica independently and proactively.

**Acknowledgements** The authors gratefully acknowledge the support of study participants and the host NHS Trust.

**Contributors** All authors contributed to the conception of this paper. CR and LR contributed to the conception and design of the study. CR performed data collection. CR, CJP and LR contributed to material preparation and analysis. Each draft of the manuscript was written by CR and commented on by CJP and LR. All authors read and approved the final manuscript.

**Funding** This research and preparation of the manuscript was funded by Health Education England (HEE) and the National Institute of Health Research (NIHR) as part of a Master's degree in Clinical Research and a Clinical Doctoral Research Fellowship (Round 4), awarded to CR, both undertaken at the University of Southampton. LR was funded, in part, by a NIHR Senior Clinical Lecturer award (Round 3).

**Disclaimer** This paper presents independent research funded by the National Institute for Health Research (NIHR). The views expressed are those of the author(s) and not necessarily those of the NHS, the NIHR or the Department of Health and Social Care.

**Competing interests** None declared.

**Patient consent for publication** Not required.

**Ethics approval** Ethical approval was gained from the South-West Ethics Committee in September 2015 (15/SW/0247).

**Provenance and peer review** Not commissioned; externally peer reviewed.

**Data availability statement** Data are available upon reasonable request. The data sets generated during and/or analysed during the current study are not publicly available due to a restriction within the ethical agreement but are available from the corresponding author on reasonable request.

**ORCID iD**
Clare Ryan http://orcid.org/0000-0002-3555-8624

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
