## [Reviewer comments · BMJ Open]

ARTICLE DETAILS

TITLE (PROVISIONAL)	Why managing sciatica is difficult: patients' experiences of an NHS sciatica pathway. A qualitative interpretative study.
AUTHORS	Ryan, Clare; Pope, Catherine; Roberts, Lisa

VERSION 1 – REVIEW

REVIEWER	Ashley Cole Sheffield Children's and Northern General Hospitals Sheffield UK
REVIEW RETURNED	02-Feb-2020

GENERAL COMMENTS	This is a very detailed qualitative review of patients with a clinical diagnosis of radicular symptoms at the time they have received the results of their MRI scans. I am not an expert in qualitative work although have been previously involved in such work. I will therefore concentrate on the clinical aspects of the work and the conclusions related to the National Back and Radicular Pain Pathway (NBRPP). 1. It seems that the NBRPP has only been implemented relatively recently in the area concerned although this is not specified. It would be useful to have some background as to how long the new Pathway has been implemented and the nature of the previous system - waiting time for patients to be seen by a Spinal Consultant, waiting lists for surgery etc.2. p4 line 5 - 'sciatica is a common form of low back pain'. There is much confusion regarding back pain and radicular leg pain and this adds to it. 'Back pain' is pain in the back. Sciatica are symptoms from the leg symptoms as they subsequently adequately describe. Just miss these few words out.3. When quoting the natural history of sciatica (Reference 4), this is not a good reference as Haugen et al (2011) obtained their patients from hospital clinics some way into their natural history. A better reference would be Alentado et al (2014) Global Spine Journal - Optimal Duration of Conservative Management Prior to Surgery for Cervical and Lumbar Radiculopathy: A Literature Review. This suggests a more realistic figure of only 15% with significant radicular symptoms at 12 weeks. This is much more supportive of an early conservative approach. This is a good time to say, I agree with your implications that management options should remain clear to patients and there needs to be more information resources.4. It would be useful for the reader if the authors in the introduction document the recommended time line in the NBRPP – 2 week review after initial presentation; for radicular pain referral at 2-6 weeks if severe pain or at 6 weeks if non-tolerable pain; MRI to be done within 4 weeks and review for results within 2 weeks of MRI.
--

	Where imaging supports the suspected clinical diagnosis of sciatica then referral for injection or surgical opinion. The NBRPP has 'shared decision making/patient choice' in every 'box'. 5. The word 'agency' and 'agent' are used 25 and 5 times respectively in the paper with no definition. This paper will be of most use to those involved in the Pathway from GP to Physio to Spinal Consultant and the terminology should reflect this. 6. Figure 1 seems to end prematurely. 7. At no stage is there any suggestion that the delivery of the NBRPP may be the relevant issue here. A similar study in a more established pathway may result in very different findings. I note Janet age 73 had 7 months of pain, had failed conservative management and had concordant imaging and was referred for a nerve root block to pain management with a waiting list of 18 weeks. I am not too surprised that this resulted in some patient disappointment. In summary, I think this is a good study and learning from patient feedback and using this to improve a service is vital in the NHS. However, I think the conclusions fail to recognise the real findings of this study and would like to see a recommendation of:  1. Their service working on the perceived problems of GP education, time-line issues, joined up services, access to injections, pain management and possibly surgical review 2. Improved patient information about the Pathway and the rationale behind each step – if they understand why, they are likely to feel more in control. 3. A suggestion that their methods are repeated in a similar service that has been in place longer. And a repeat of this in their own service in a couple of years to see if there have been any changes.
--	---

REVIEWER	Nefyn Williams University of Liverpool UK
REVIEW RETURNED	11-Feb-2020

GENERAL COMMENTS	This is an interesting qualitative study that deals with an important problem. The qualitative methods are robust. I have only minor comments:  1. There are often long sentences with many clauses, especially in the Discussion. These could be split into separate sentences to improve clarity. For, example lines 53-58 on the 1st page of the discussion, lines 3-6 and lines 16-18 on the 2nd page of the discussion. 2. Article summary, strengths and limitations, 3rd bullet point and also 1st page discussion line 29 "these issues occurred as the methods were determined by the requirements of the original study" Please clarify. 3. Introduction - Briefly discuss the overlap with referred leg pain. 4. Introduction line 35 - cross sectors better than cross traditional boundaries? 5. Discussion - Also discuss the contrary view about the value of stepped care for reducing the need for invasive treatment and also being more cost-effective (Fitzpatrick). 6. Typos, grammatical suggestions 2nd page of method line 51 episode of back pain (to minimise bias). End of results line 7 to access private treatment,
--

	Results Theme 3 Agentic burden - I could not find agentic in Collins dictionary. What about "Burden of agency" instead? Also 1st page of discussion on line 32 and Figure 3 7. Figure 1 is incomplete Reference Fitzsimmons D, Phillips CJ, Bennett H, Jones M, Williams NH, Lewis R, Sutton A, Matar HE, Din NU, Burton K, Nafees S, Hendry M, Rickard I, Wilkinson C. Cost effectiveness of different management strategies for sciatica. Pain 2014; 155: 1318-1327.
--	--

VERSION 1 – AUTHOR RESPONSE

Response to reviewers' recommendations for revision.			
Manuscript ID bmjopen-2020-037157 entitled "Why managing sciatica is difficult: patients' experiences of an NHS sciatica pathway. A qualitative interpretative study."			
Com ment no.	Comment	Page and paragraph no. (in clean copy)	Reply and revision
Reviewer 1 Ashley Cole			
0	This is a very detailed qualitative review of patients with a clinical diagnosis of radicular symptoms at the time they have received the results of their MRI scans. I am not an expert in qualitative work although have been previously involved in such work. I will therefore concentrate on the clinical aspects of the work and the conclusions related to the National Back and Radicular Pain Pathway (NBRPP).	N/A	Thank you for such a helpful review.
1	It seems that the NBRPP has only been implemented relatively recently in the area concerned although this is not specified. It would be useful to have some background as to how long the new Pathway has been	P4 under the heading 'Setting'	Following reflecting on this point and comment 7, we recognise that the study explores patients' experiences of the local pathway, which was working to align with national pathway recommendations. We have therefore amended the title to: 'Why managing sciatica is difficult:

	implemented and the nature of the previous system - waiting time for patients to be seen by a Spinal Consultant, waiting lists for surgery etc.		patients' experiences of an NHS sciatica pathway. A qualitative, interpretative study'. In addition, the following details have been added: 'At the time data was collected, the first version of the national pathway had been available for 16 months. Over this period changes had occurred in the spinal surgery, injection and imaging providers. Not all providers were commissioned and funded by the same organisation. Issues with staff shortages and provider negotiations had resulted in waiting times of 6 months for a routine appointment with a spinal surgeon and 4 months for an injection (however, patients could choose to access this in the adjacent city within 4 weeks). Following a recent change in provider, the wait for imaging had reduced from 6 to 3 weeks.'
2	'sciatica is a common form of low back pain'. There is much confusion regarding back pain and radicular leg pain and this adds to it. 'Back pain' is pain in the back. Sciatica are symptoms from the leg symptoms as they subsequently adequately describe. Just miss these few words out.	P3 line 4	This section now reads: 'Sciatica is characterised by pain, altered sensation and/or weakness in the leg.'
3	When quoting the natural history of sciatica (Reference 4), this is not a good reference as Haugen et al (2011) obtained their patients from hospital clinics some way into their natural history. A better reference would be Alentado et al (2014) Global Spine Journal - Optimal Duration of Conservative Management Prior to Surgery for Cervical and Lumbar	P3 paragraph 1	Thank you for the reference. Our understanding is that the proportion of patients who have ongoing symptoms at 3 months and beyond is not yet well understood because different studies have different findings. Whilst this may well relate to the duration of symptoms at the time of presentation, other associated factors might also explain the variation. Also, the current situation in primary care is that many patients do present with a several month history of

	Radiculopathy: A Literature Review. This suggests a more realistic figure of only 15% with significant radicular symptoms at 12 weeks. This is much more supportive of an early conservative approach. This is a good time to say, I agree with your implications that management options should remain clear to patients and there needs to be more information resources.		symptoms. We think it is important for readers to understand that there is a group of patients with persistent sciatic symptoms. This section now reads: 'Whilst the majority of patients with sciatica are likely to experience early improvement in symptoms, usually in the first 2-3 months, either with or without treatment; a minority will, experience more persistent symptoms or disability, and for some this continues beyond 12 months. Also, some patients will experience intermittent or recurrent sciatic symptoms over time. There is, however, inconsistency in the literature about the proportion of patients affected by ongoing symptoms. Whilst one review found this to be as few as 13% of patients at 6 months(6), another study based in primary care, found that for 45% of patients, disability had failed to significantly improve at 12 months(7). The reason for this variation is not fully understood but may be associated with symptom duration at presentation or the type of studies to which patients were recruited (clinical versus epidemiological). Recent work indicates that factors negatively associated with recovery include longer leg pain duration; more symptoms associated with sciatica; and patient's belief that the problem would last a long time; conversely, having myotomal weakness was positively associated with recovery(7).'
4	It would be useful for the reader if the authors in the introduction document the recommended time line in the NBRPP – 2 week review after initial presentation; for radicular pain referral at 2-6 weeks if severe pain or at 6 weeks if non-tolerable pain; MRI to be done within 4 weeks and review for results within 2 weeks of MRI.	P3 paragraph 2	This section now reads 'The new national pathway recommends that patients with sciatica are initially managed within primary care, with medication and/or physiotherapy, with a 2-week review after initial presentation. For those with severe pain that fails to improve, assessment by a specialist (spinal triage) practitioner is recommended at 2-6 weeks, or at 6 weeks for those with non-tolerable pain. In the UK this

	Where imaging supports the suspected clinical diagnosis of sciatica then referral for injection or surgical opinion. The NBRPP has 'shared decision making/patient choice' in every 'box'.		specialist role is commonly performed by an advanced practice physiotherapist(10). The pathway recommends that investigations such as magnetic resonance imaging (MRI) are offered (if they are likely to change management) and completed within 4 weeks, with results review within 2 weeks. Where imaging supports the suspected clinical diagnosis of sciatica, referral for an epidural injection or surgery is advised, with surgery occurring within 8-12 weeks of the onset of symptoms, or for those with very severe symptoms, within 3 weeks. When no concordant structural cause is identified, or the patient wishes to avoid invasive intervention, the recommended approach is a combined physical and psychological programme. The pathway advocates that clinicians share decision making with patients at each stage of the pathway.'
5	The word 'agency' and 'agentic burden' are used 25 and 5 times respectively in the paper with no definition. This paper will be of most use to those involved in the Pathway from GP to Physio to Spinal Consultant and the terminology should reflect this.	N/A	The term agency has now been defined in the second paragraph of the results section. This reads: 'The term 'agency' refers to the independent and proactive actions undertaken by a patient as part of the self-management of their health(21).' We have replaced the term 'agentic burden' with 'the burden of agency'. We have reduced the number of times the word agency is used to 12, replacing it with the term 'independent and proactive' where possible to aid understanding. We do however think that it is important that readers are encouraged to recognise that the findings align with the concepts of agency and treatment burden as this helps identify their potential relevance to similar services.

6	Figure 1 seems to end prematurely.	Figure document	The text box has now been expanded to fit contents.
7	At no stage is there any suggestion that the delivery of the NBRPP may be the relevant issue here. I similar study in a more established pathway may result in very different findings. I note Janet age 73 had 7 months of pain, had failed conservative management and had concordant imaging and was referred for a nerve root block to pain management with a waiting list of 18 weeks. I am not too surprised that this resulted in some patient disappointment.	Title, abstract P2 L24 & Discussion P10 L60; P11 L9, 24	Following reflecting on this point (and comment 1) we recognise that the study explores patients' experiences of the local pathway, which was working to align with national pathway recommendations. We have amended the title to: 'Why managing sciatica is difficult: patients' experiences of an NHS sciatica pathway. A qualitative, interpretative study'. We have also used the phrase 'local pathway' rather than pathway to indicate where the findings specifically relate to local implementation. Finally, we have added the following sentences to the second paragraph of the discussion: 'We suggest that they may have arisen because the local pathway focused on delivering the 'mechanics' of the pathway (procuring services; enabling faster access to imaging; and setting up regular case discussion meetings) and did not prioritise ensuring that management was person centred. Our data reveals a culture of paternalism, protocol-driven care, information being provided on a 'need to know basis', and siloed services. Patients' frustration with the lack of timely access to a specialist opinion, likely reflects in part the known issues with staff shortages and difficulty negotiating provider contracts. However, in addition, the national pathway places a new emphasis on enabling rapid access to imaging and injection or surgery (for those with severe or intolerable symptoms and a relevant cause identified on imaging) and achieving this requires a shift in practice and resources. Whilst this approach reflects expert opinion, it is in contrast to the established 'stepped model' of sciatica care that advocates the more limited use of invasive interventions, with evidenced cost effectiveness(27). We

			suggest that until the national pathway is underpinned by evidence of its clinical and cost effectiveness, services may be ambivalent about making the changes necessary to align with its timelines. Whilst these issues are specific to the local pathway, we suggest their conceptual transferability to other services in the early stages of pathway implementation.'
8	In summary, I think this is a good study and learning from patient feedback and using this to improve a service is vital in the NHS. However, I think the conclusions fail to recognise the real findings of this study and would like to see a recommendation of:  1. Their service working on the perceived problems of GP education, time-line issues, joined up services, access to injections, pain management and possibly surgical review 2. Improved patient information about the Pathway and the rationale behind each step – if they understand why, they are likely to feel more in control. 3. A suggestion that their methods are repeated in a similar service that has been in place longer. And a repeat of this in their own service in a couple of years to see if there have been any changes. 	Implications P11 L49	Thank you. The implications sections now reads: 'Local implications include the need to improve and then re-evaluate the following aspects of care: patient information about the pathway; staff training and patient resources to enable effective shared decision making; collaborative, joined up working; and timely progression through the pathway. Also, patients' experiences should be routinely explored as part of service evaluation. Transferable implications include the need to ensure pathway care is person-centred and provides the support and resources necessary to enable and empower patients to manage sciatica independently and proactively. Further research exploring patients' experiences of the new national pathway, particularly in services in which the new pathway is fully implemented, would enable a more comprehensive understanding of patients' experience of being managed in NHS sciatica pathways.'
Reviewer 2: Nefyn Williams			
0	This is an interesting qualitative study that deals with an important problem. The qualitative methods are robust. I have only minor comments:		Thank you for such a helpful review.

1	There are often long sentences with many clauses, especially in the Discussion. These could be split into separate sentences to improve clarity. For, example lines 53-58 on the 1st page of the discussion, lines 3-6 and lines 16-18 on the 2nd page of the discussion.	Discussion P11 L32 & L38	These sentences have now been shortened and simplified. They now read: 'This paper furthers understanding by reframing patients' actions as agency, and arguing that this agency becomes necessary in part due to the problems with the pathway as well as in response to wider neo-liberal expectations that people manage their own health and reduce dependence on the state (28). This study shows that it has become normal for both patients and staff to expect patients to be proactive and independent in their management.' 'However, our study also found that the requirement to be independent and proactive when managing sciatica was difficult and burdensome to patients. This was because of the impact of sciatica and patients' lack of skill, funds and support. The Burden of Treatment Theory recognises that when patients are expected to participate in their healthcare but lack adequate support, this adversely affects their ability to manage their illness (29). Our analysis demonstrates the particular burden of being required to navigate the pathway issues and to independently and proactively manage sciatica without the necessary support.'; The final sentence mentioned (first line of the implications) has now been removed in response to reviewer 1's comment 8.
2	Article summary, strengths and limitations, 3rd bullet point and also 1st page discussion line 29 "these issues occurred as the methods were determined by the requirements of the original study" Please clarify.	Article summary, strengths and limitations P2 L55 and Discussion P10 L52	This section now reads: 'Limitations include not undertaking iterative theoretical sampling during analysis and interviewing some participants before they had completed their management'. The reasons for the limitations are now explained in the discussion and this now reads: 'Limitations include not undertaking iterative theoretical sampling during analysis (because

			analysis for this paper was completed after data collection had ended) and interviewing some participants before they had completed their management (to align with the aim of the original study, patients were interviewed within 6 weeks of undergoing investigations).'
3	Introduction - Briefly discuss the overlap with referred leg pain.	Introduction P3 L8	The following sentence has been added to clarify the distinction between sciatica and somatic referred leg pain: 'Sciatica differs from somatic referred leg pain which is attributed to structures other than the nerve root such as the joints, muscles and ligaments.'
4	Introduction line 35 - cross sectors better than cross traditional boundaries?	Introduction section P3 L38	As suggested, 'cross traditional boundaries of care' has been substituted with 'cross sectors of care'.
5	Discussion - Also discuss the contrary view about the value of stepped care for reducing the need for invasive treatment and also being more cost-effective (Fitzsimmons D, Phillips CJ, Bennett H, Jones M, Williams NH, Lewis R, Sutton A, Matar HE, Din NU, Burton K, Nafees S, Hendry M, Rickard I, Wilkinson C. Cost effectiveness of different management strategies for sciatica. Pain 2014; 155: 1318-1327.)	Discussion P11 L13	Thank you. To address this, we have added the following information to the second paragraph of the discussion: 'Patients' frustration with the lack of timely access to a specialist opinion, likely reflects in part the known issues with staff shortages and difficulty negotiating provider contracts. However, in addition, the national pathway places a new emphasis on enabling rapid access to imaging and injection or surgery (for those with severe or intolerable symptoms and a relevant cause identified on imaging) and achieving this requires a shift in practice and resources. Whilst this approach reflects expert opinion, it is in contrast to the established 'stepped model' of sciatica care that advocates the more limited use of invasive interventions, with evidenced cost effectiveness(27). We suggest that until the national pathway is underpinned by evidence of its clinical and cost effectiveness, services may be ambivalent about making the changes necessary to align with its

			timelines. Whilst these issues are specific to the local pathway, we suggest their conceptual transferability to other services in the early stages of pathway implementation.'
6	Typos, grammatical suggestions (i) 2nd page of method line 51 episode of back pain (to minimise bias). (ii) End of results line 7 to access private treatment, (iii) Results Theme 3 Agentic burden - I could not find agentic in Collins dictionary. What about "Burden of agency" instead? Also 1st page of discussion on line 32 and (iv) Figure 3 7. Figure 1 is incomplete	P5 L19 P10 L34 P6 L31, P10 L13 Figures document	Thank you. These sections now read: (i) '(to minimise bias)'. (ii) 'Some reported lacking the financial resources to access to private treatment, especially if multiple treatments, or costly interventions such as injections or surgery, were needed.' (iii) As suggested, theme 3 has been changed throughout the document to 'the burden of agency'. (iv) The text box has been expanded to fit the contents.
Formatting amendments			
1	Required format for each figure Figures can be supplied in TIFF, JPG or PDF format (figures in document, excel or powerpoint format will not be accepted), we also request that they have a resolution of at least 300 dpi and 90mm x 90mm of width.	N/A	The documents have now been submitted in the required format.
2	We have noticed that you have uploaded the file "Coding Tree.docx (v1.0)" under 'supplementary file'. However, we cannot see any citation for this file within the main text. If this file needs to be published as supplementary file, please cite it as 'supplementary file' in the main text.	N/A	The designation of this file has been changed to 'Supplementary file for editors only'.

	Otherwise, kindly change its file designation to 'Supplementary file for editors only'.		
--	---	--	--

VERSION 2 – REVIEW

REVIEWER	Ashley Cole Sheffield Children's Hospital England
REVIEW RETURNED	14-Mar-2020

GENERAL COMMENTS	The authors have dealt with all the concerns I raised in my original review
---

REVIEWER	Nefyn Williams University of Liverpool, UK I was co-author of a recent qualitative research paper discussed in this article.
REVIEW RETURNED	05-Mar-2020

GENERAL COMMENTS	I am happy with the authors' responses
--